# Har-P, a short P-element variant, weaponizes P-transposase to severely impair Drosophila development

Satyam P Srivastav[1], Reazur Rahman[2], Qicheng Ma[1], Jasmine Pierre[1], Saptaparni Bandyopadhyay[1], Nelson C Lau[1,2,3]*

[1]Department of Biochemistry, Boston University School of Medicine, Boston University, Boston, United States; [2]Department of Biology, Brandeis University, Waltham, United States; [3]Genome Science Institute, Boston University School of Medicine, Boston, United States

**Abstract** Without transposon-silencing Piwi-interacting RNAs (piRNAs), transposition causes an ovarian atrophy syndrome in Drosophila called gonadal dysgenesis (GD). Harwich (Har) strains with P-elements cause severe GD in F1 daughters when Har fathers mate with mothers lacking P-element-piRNAs (i.e. ISO1 strain). To address the mystery of why Har induces severe GD, we bred hybrid Drosophila with Har genomic fragments into the ISO1 background to create HISR-D or HISR-N lines that still cause Dysgenesis or are Non-dysgenic, respectively. In these lines, we discovered a highly truncated P-element variant we named 'Har-P' as the most frequent de novo insertion. Although HISR-D lines still contain full-length P-elements, HISR-N lines lost functional P-transposase but retained Har-P's that when crossed back to P-transposase restores GD induction. Finally, we uncovered P-element-piRNA-directed repression on Har-P's transmitted paternally to suppress somatic transposition. The Drosophila short Har-P's and full-length P-elements relationship parallels the MITEs/DNA-transposase in plants and SINEs/LINEs in mammals.

*For correspondence:
nclau@bu.edu

## Introduction

The sterility syndrome of 'P-M' hybrid dysgenesis in Drosophila melanogaster (*Engels and Preston, 1979*; *Kidwell et al., 1977*) is due to uncontrolled P-element transposition that damages ovarian development and induces female sterility (*Bingham et al., 1982* and reviewed in *Kelleher, 2016*). This gonadal dysgenesis (GD) phenotype occurs in hybrid F1 daughters whose paternal genome comes from a father possessing active P-elements (a 'P' strain) and a maternal genome unable to express P-element piRNAs (an 'M' strain) (*Brennecke et al., 2008*; *Khurana et al., 2011*). The fascinating nature of this genetic syndrome is complete fertility in daughters from the reciprocal cross because the mother possessing active P-elements contribute P-element-derived Piwi-interacting RNAs (piRNAs) to silence these transposons in daughters (*Brennecke et al., 2008*; *Khurana et al., 2011*). Thus, despite identical genetic makeup in daughters between the reciprocal crosses, the epigenetic maternal transmission of transposon repression by piRNAs starkly defines female fertility (illustrated in *Figure 1A*).

Between fertility and complete sterility lies a spectrum of GD induction variation amongst different strain crosses that may be attributed to differential P-element copy numbers in different strain genomes (*Anxolabéhère et al., 1985*; *Bergman et al., 2017*; *Biémont et al., 1990*; *Bingham et al., 1982*; *Boussy et al., 1988*; *Kidwell et al., 1981*; *Ronsseray et al., 1989*; *Srivastav and Kelleher, 2017*; *Yoshitake et al., 2018*), and capacity to generate piRNAs (*Wakisaka et al., 2017*). In addition, there are many non-autonomous P-element variants that can be mobilized by P-transposases, including very short elements from the pi[2] strain (*Bingham et al., 1982*; *O'Hare and Rubin, 1983*)

**eLife digest** DNA provides the instructions needed for life, a role that relies on it being a very stable and organized molecule. However, some sections of DNA are able to move from one place in the genome to another. When these "mobile genetic elements" move they may disrupt other genes and cause disease. For example, a mobile section of DNA known as the *P*-element causes a condition called gonadal dysgenesis in female fruit flies, leading to infertility.

Only certain strains of fruit flies carry *P*-elements and the severity of gonadal dysgenesis in their daughters varies. For example, when male fruit flies of a strain known as *Harwich* (or *Har* for short) is crossed with female fruit flies that do not contain *P*-elements, all of their daughters develop severe gonadal dysgenesis and are infertile. However, if the cross is done the other way around, and female *Har* flies mate with males that do not contain *P*-elements, the daughters are fertile because the *Har* mothers provide their daughters with protective molecules that silence the *P*-elements. But it was a mystery as to why the *P*-elements from the *Har* fathers always caused such severe gonadal dysgenesis in all the daughters.

Here, Srivastav et al. bred fruit flies to create offspring that had different pieces of *Har* DNA in a genetic background that was normally free from *P*-elements; they then analyzed the 'hybrid' offspring to identify which pieces of the *Har* genome caused gonadal dysgenesis in the daughter flies. These experiments showed that *Har* flies possess a very short variant of the *P*-element (named "*Har-P*") that is more mobile than other variants. However, the *Har-P* variants still depended on an enzyme known as *P*-transposase encoded by the full-length *P*-elements to move around the genome. Further experiments showed that other strains of fruit flies that cause severe gonadal dysgenesis also had very short *P*-element variants that were almost identical to *Har-P*.

These findings may explain why *Har* and some other strains of fruit flies drive severe gonadal dysgenesis. In the future, it may be possible to transfer *P*-transposase and *Har-P* into mosquitoes, ticks and other biting insects to make them infertile and help reduce the spread of certain diseases in humans.

that actually assemble in vitro with the *P*-transposase tetramer complex >100X more efficiently than the full-length *P*-element (*Tang et al., 2007*). However, many earlier studies perceived truncated variants such as the '*KP2*' variants as inhibitors of transposition by acting to titrate *P*-transposase since *P*-element piRNAs were unknown at the time (*Black et al., 1987*; *Gloor et al., 1993*; *Jackson et al., 1988*; *Robertson and Engels, 1989*; *Simmons et al., 2002a*). Most studies of GD were typically calibrated with a strong paternal inducer '*P*'-strain like *Harwich* (*Har*) or *pi[2]* when mated with '*M*' strain females lacking *P*-elements (*Bingham et al., 1982*; *Brennecke et al., 2008*; *Kidwell et al., 1977*; *Rubin et al., 1982*). Despite over 40 years of study, what defines a strong paternal inducer of GD has remained a mystery.

Although *P*-element copy numbers in *Har* are significant (120–140 copies; *Khurana et al., 2011*), strains with even more copies like *OreR-MOD* do not induce GD whereas other strong inducer strains that have >75% fewer *P*-element copies than *Har* can also trigger complete GD (*Figure 1B and C*). Thus, there is a lack of correlation between *P*-element copy number and GD induction (*Figure 1D*) that we and others have previously observed (*Bergman et al., 2017*; *Ronsseray et al., 1989*; *Srivastav and Kelleher, 2017*). Since *P*-element copy numbers do not explain GD severity, we hypothesized that a special *P*-element variant or insertion locus might underlie the strong GD phenotype in certain strong '*P*' strains like *Har*. To discover this *P*-element variant, we undertook a reductionist approach to find specific *P*-element variant(s) required for GD induction that revealed unexpectedly a short variant from the *Har* strain that may act together with the full-length *P*-transposase to drive strong GD.

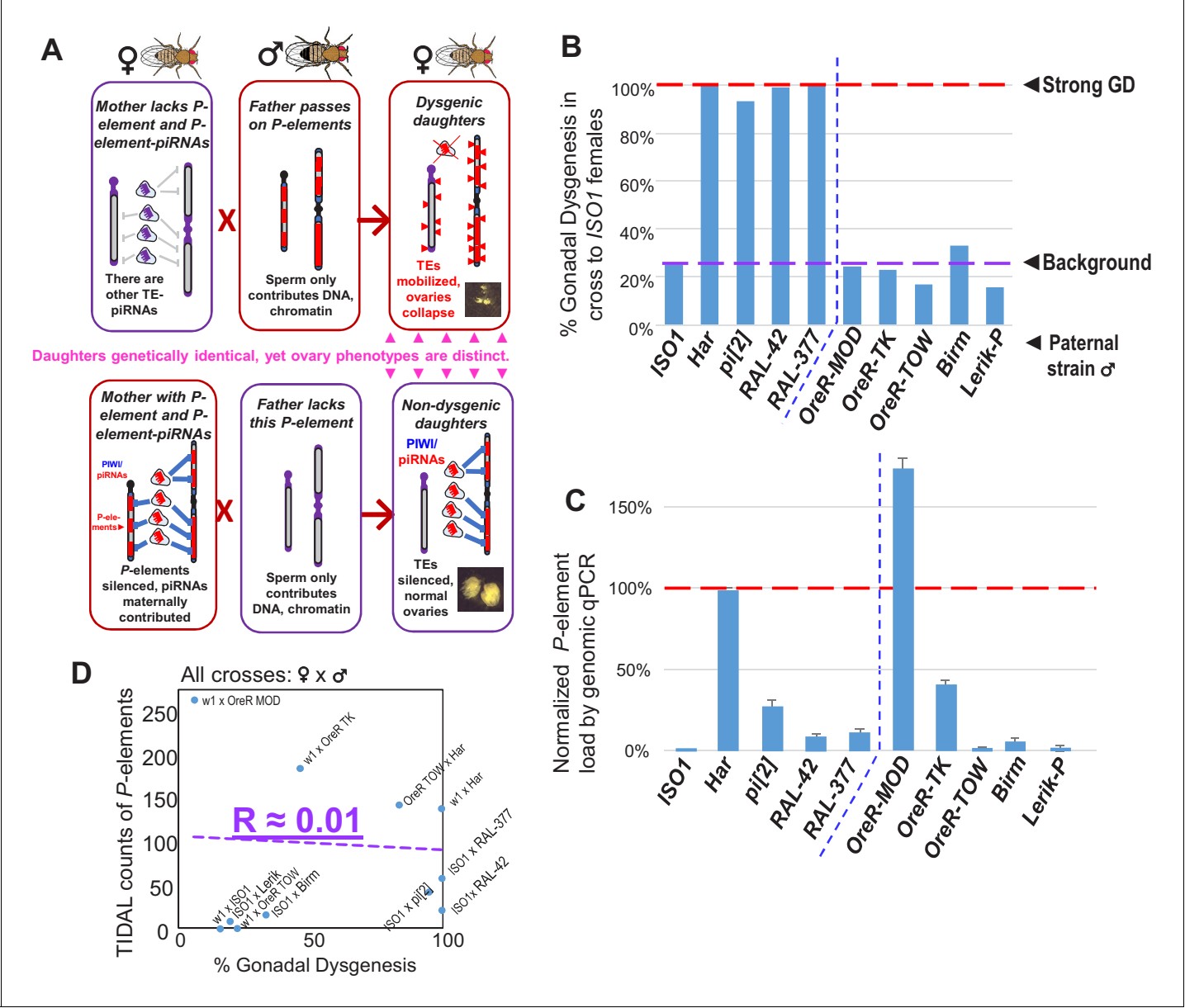

**Figure 1.** No correlation between paternally-induced gonadal dysgenesis (GD) rate and *P*-element copy number. (**A**) Illustration of the *P*-element-induced GD phenomenon, where two different types of crosses with one parent lacking *P*-elements while the other parents containing *P*-elements can result in genetically identical daughters having very different gonadal phenotypes. (**B**) GD rates from paternal genome strains mated with *ISO1* females; at least 100 F1 daughters per cross were assayed. (**C**) Genomic quantitative PCR assessment of *P*-element load of strains, normalized to *Har* at 100%. (**D**) Scatterplot comparing TIDAL counts of *P*-element insertions to the GD rate reflects the lack of correlation.

The online version of this article includes the following source data for figure 1:

**Source data 1.** Spreadsheets with the tabulation of gonadal dysgenesis assays and raw values from the qPCR experiments.

## Results

### GD retention and loss in hybrid *Drosophila* lines with reduced *P*-element copy numbers

To genetically isolate the causative transposon strongly inducing GD and facilitate discovery by whole genome sequencing (WGS), we generated hybrid lines where only a minor fraction of the *Har* genome is within the background of the *ISO1* reference genome sequence. We first conducted several fertility-permissive backcrosses between female *Har* and male *ISO1*, selecting hybrid progeny

that propagated a red-eye phenotype which we attributed to the 'red' eyes due to *Har* alleles replacing the *cn*, *bw*, *sp*, alleles on Chromosome 2R (Chr2R) of *ISO1* (***Figure 2A***-abridged scheme, ***Figure 2—figure supplement 1***-detailed scheme). We then performed an initial GD validation screen with many vials of individual hybrid males crossed to *ISO1* females and selecting for lines that caused 100% GD from this cross. Lines were propagated with additional self-crosses and further inbred with single-sibling pairs. We then subjected multiple independent *Har-ISO1-Selfed-Red* (*HISR*) lines to a second GD assay. Finally, we conducted qPCR to identify the lines with the greatest reduction of *P*-element copy numbers (***Figure 2B***) and settled on four lines each that either retained severe paternally-induced GD (*HISR-D*) or had lost this capacity (*HISR-N*) (***Figure 2C***).

Genomic PCR genotyping of deletion loci of *Har* compared to *ISO1* in *HISR* lines indicated that these lines carried mostly *ISO1* genomes (***Figure 2—figure supplement 2***). Therefore, we

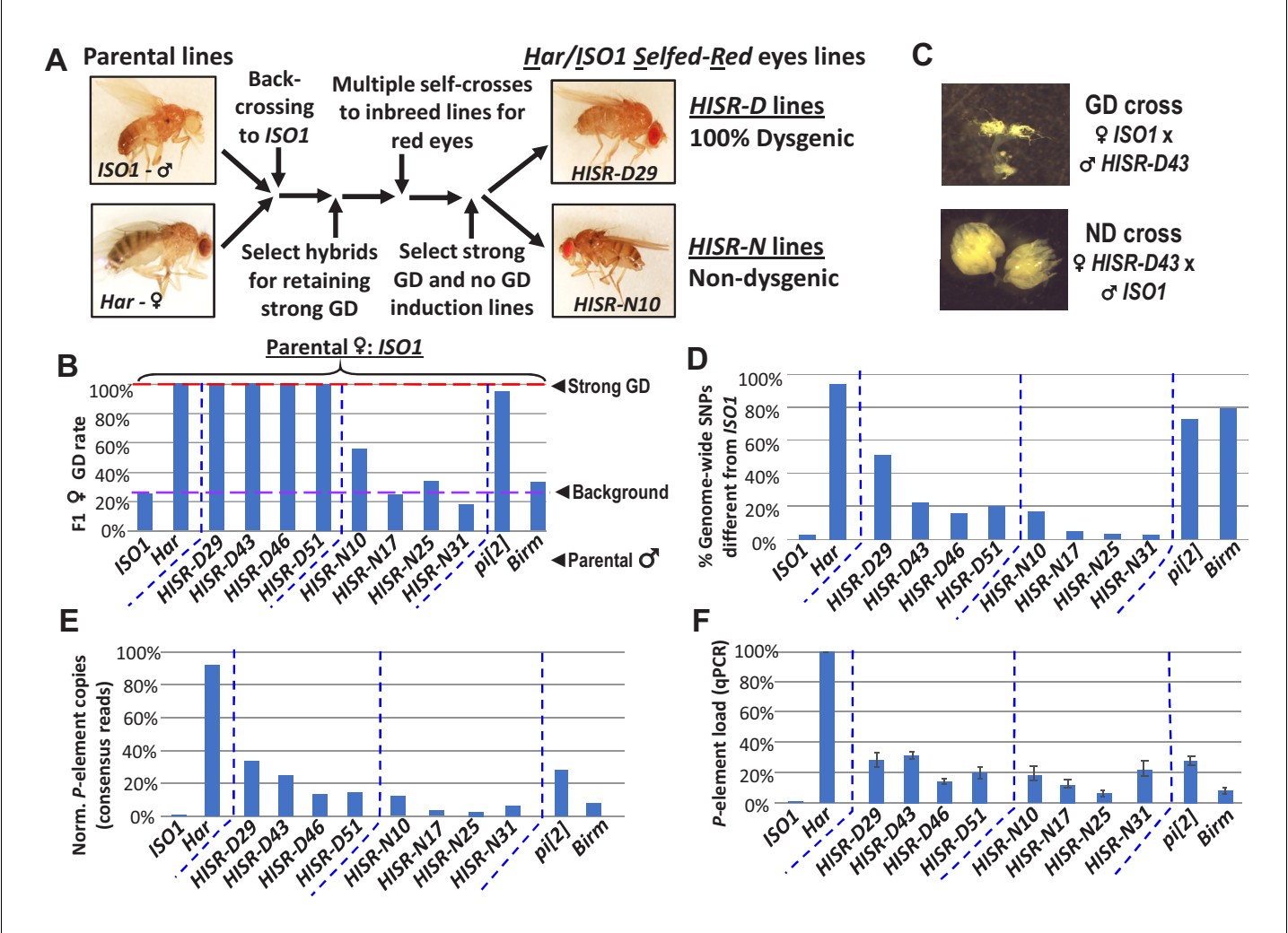

**Figure 2.** *HISR* lines retaining or losing strong gonadal dysgenesis (GD) induction. (**A**) Abridged scheme for generating hybrid lines retaining a small fraction of the *Har* genome in the *ISO1* background. Full scheme is ***Figure 2—figure supplement 1***. (**B**) GD rates from paternal genome lines and strains mated with *ISO1* females. (**C**) Ovarian atrophy phenotype only observed from paternal induction of GD. (**D**) Genome-wide single nucleotide polymorphism profile differences distinct from *ISO1* genome. (**E**) Normalized counts of *P*-element copies by consensus read mapping of genomic libraries. (**F**) qPCR assessment of total genomic *P*-element load.

The online version of this article includes the following source data and figure supplement(s) for figure 2:

**Source data 1.** Spreadsheets with the tabulation of gonadal dysgenesis assays, quantitation of genomewide SNP profiles, calculations of P-element genomic loads, and raw values from the qPCR experiments.
**Figure supplement 1.** Detailed scheme for generating hybrid strains of reduced *Har* genome within the *ISO1* background.
**Figure supplement 2.** Genomic PCR genotyping of *HISR* strains.

performed WGS of the parental *Har* and *ISO1* strains, the 8 *HISR* lines, and the *pi[2]* and *Birmingham* (*Birm*) strains, two classic strains with similar numbers of *P*-elements but diametric capacity to induce GD (*Engels et al., 1987*; *Simmons et al., 1987*). Single-nucleotide polymorphism (SNP) profiles of *HISR* lines confirmed that only a small percentage of the *Har* genome was retained in mostly an *ISO1* background (*Figure 2D*). Quantification of *P*-element copies from WGS with the TIDAL program (Transposon Insertion and Depletion AnaLyzer, *Figure 2E*) (*Rahman et al., 2015*) was also consistent with qPCR measurements (*Figure 2F*).

## *HISR-D* lines produce similar levels of *P*-element piRNAs as the parental *Har* strain

To determine how substantial reduction in *P*-element copy numbers in *HISR* lines affected *P*-element-directed piRNA production, we generated and sequenced highly-consistent ovarian small RNA libraries (*Figure 3A*) and confirmed the expected presence and absence of *P*-element piRNAs in *Har* and *ISO1* ovaries, respectively (*Figure 3B*). Surprisingly, there were similar-to-increased levels of *P*-

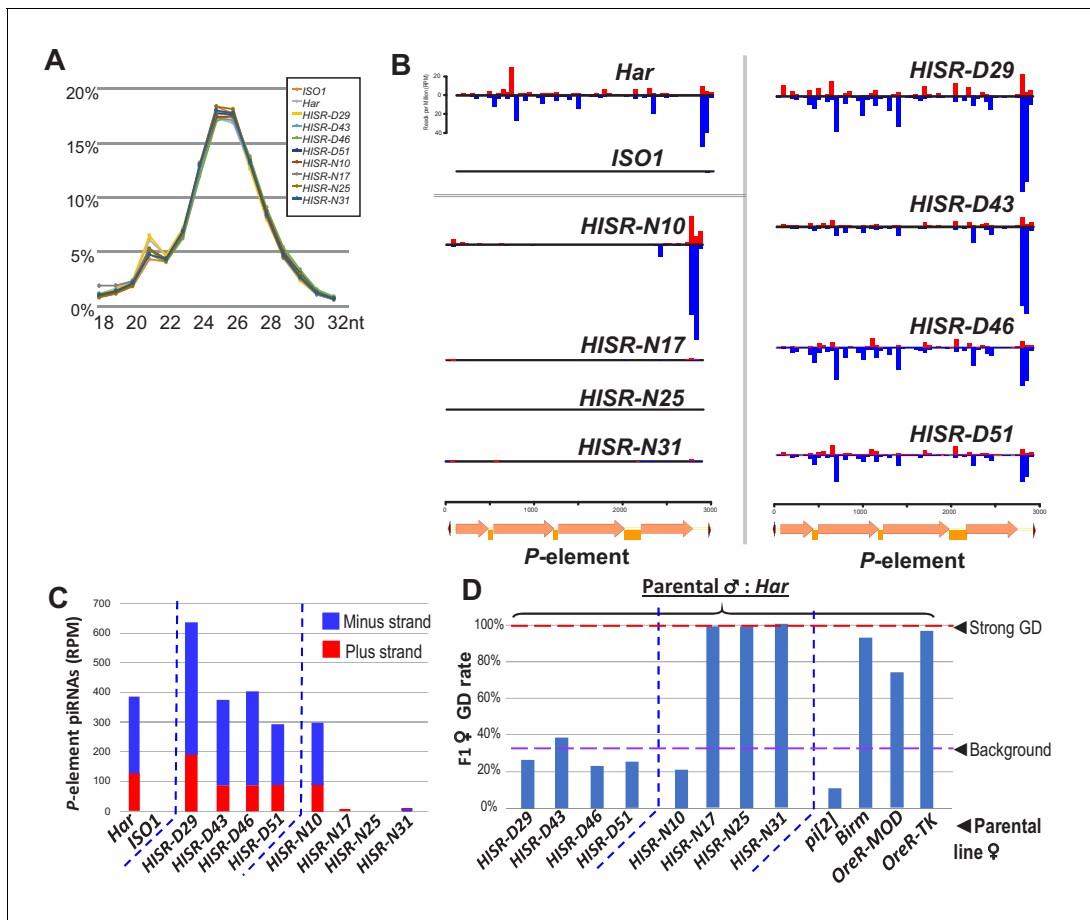

**Figure 3.** *P*-element directed piRNA production in *HISR* strains ovaries. (A) Nearly identical size distributions of small RNAs from parental and *HISR* ovaries. (B) *P*-element piRNAs coverage plots and (C) quantitation of the *P*-element piRNAs mapping to plus and minus strands, in reads per million (RPM). (D) Assays for repression of *P*-element induced GD for *HISR* strains (N ≥ 100 females) are a good proxy for production of piRNAs silencing *P*-elements.

The online version of this article includes the following source data and figure supplement(s) for figure 3:

**Source data 1.** Spreadsheets with the calculations for P-element piRNA coverage and tabulation of gonadal dysgenesis assays.

**Figure supplement 1.** Candidate locus for *P*-element directed piRNA production in *HISR-N10* ovaries.

**Figure supplement 1—source data 1.** Spreadsheet with the tabulation of gonadal dysgenesis assays.

**Figure supplement 2.** Analysis of piRNA ping-pong signatures on the *P-element* versus ovarian total small RNAs and specific transposons and somatic piRNA cluster loci.

element piRNAs between *HISR-D* and *Har* strains, whereas amongst the *HISR-N* lines, only *HISR-N10* retained *P*-element piRNAs (*Figure 3C*). Our own mapping analysis indicated a common 3' end anti-sense bias of *P*-element piRNAs that we also confirmed with an independent piRNA analysis pipeline (*Han et al., 2015*). These mapping patterns are consistent with piRNAs silencing transposons and suppressing hybrid GD (*Brennecke et al., 2008*; *Erwin et al., 2015*; *Khurana et al., 2011*) as well as correlating with all the *HISR-D's* and *HISR-N10's* immunity to strong GD induction when these females are mated to *Har* males (*Figure 3D*).

Additional piRNAs broadly cover the full length of *P*-element in *Har* and *HISR-D* lines (*Figure 3B*-top and right), but the notable depletion of internal *P*-element piRNAs in *HISR-N10* (*Figure 3B*-middle left) prompted us to conjecture which of its 23 TIDAL-mapped *P*-elements might be stimulating this novel piRNA pattern. We only found one euchromatic *P*-element inser-tion in *HISR-N10* that specifically coincided with an increase of local piRNAs (*Figure 3—figure supplement 1A*). This *P*-element inserted into the 5' UTR of *DIP1*, adjacent to the enhancer and promoter region of *Flamenco*, the major piRNA cluster located in a pericentromeric region of the X-chromosome (*Brennecke et al., 2007*). However, when we selected just the *HISR-N10* X-chromosome balanced with the *FM7a* balancer chromosome (*Figure 3—figure supplement 1B*), this X-chromosome locus did not generate enough *P*-element piRNAs to provide full GD immunity. It is possible for additional *P*-elements to have inserted into major piRNA cluster loci like *42AB*, *Flamenco* and *TAS*-regions as part of the endogenizing process (*Khurana et al., 2011*; *Moon et al., 2018*), but the intractable repetitiveness of piRNA cluster regions prevents bioinformatic programs from pinpointing *P*-element insertions in these regions. Interestingly, all the *P*-element piRNAs detected in *HISR-N10*, *-D29*, and parental *Har* appeared to be expressed in the germline due to the detection of clear ping-pong piRNA biogenesis signatures (*Figure 3—figure supplement 2*). Finally, the *P*-element piRNA patterns in *HISR-N10* can be explained by the abundant *P*-element variant that will be discussed below.

## Dispersed *P*-element landscapes indicate de novo transposition in *HISR* lines

The selection for 'red' eyes of *Har* alleles in *HISR* lines should have replaced the *cn, bw, sp*, alleles on Chromosome 2R (Chr2R) of *ISO1*, therefore we had hoped that WGS of *HISR* line genomes might to point to a specific set of *P*-elements responsible for inducing strong GD. Unexpectedly, the *P*-ele-ment insertions were not confined to Chr2R, but rather were dispersed across the entire genomes of all *HISR* lines (*Figure 4A*), seemingly defying the genomic PCR genotyping and WGS-SNP profiling that indicated sufficient backcrossing to favor mostly the *ISO1* genetic background (*Figure 4—fig-ure supplement 1*).

To explore this conundrum, we examined how many of the original *P*-elements in the *Har* genome were conserved in the *HISR* lines' genomes (*Figure 4B*). As expected for *HISR-D29* whose *P*-element copy numbers was closest to *Har*, this line conserved the highest share of parental *Har P*-elements compared to other *HISR* lines. However, there were also 35 novel *P*-element insertions (~45%) in *HISR-D29* absent from *Har*. Surprisingly, the vast majority of the *P*-element insertions across all *HISR* lines were also de novo P-element insertions (*Figure 4C*), with each line clearing out nearly all paren-tal *Har P*-element insertions and developing unique landscapes of *P*-element insertions. These data suggest that during the course of stabilizing the *HISR* lines, there were bursts of new *P*-element transpositions resulting in novel transposon landscapes that are completely distinct from the paren-tal *Har* genome.

Although this dispersion of de novo P-elements in *HISR* lines' genomes stymied our goal to pinpoint a particular *Har* locus strongly inducing GD, we next cloned and sequenced genomic PCR amplicons of all *P*-elements from the various *P*-element-containing strains. By using a single oligonucleotide that primes from both the 5' and 3' Terminal Inverted Repeats (TIRs), we ampli-fied full-length *P*-elements as well as several additional truncation variants (*Figure 5A*) that have been missed in other genomic PCR assays using internal primers (*Wakisaka et al., 2017*). The most abundant variant accounting for more *P*-element copies in *OreR-MOD* and *OreR-TK* strains compared to *Har* were the 'KP' variant shown to encode a dominant negative protein that inhib-its full-length *P*-transposase activity (*Jackson et al., 1988*; *Simmons et al., 1990*) (*Figure 5A–5C*), thus explaining the innocuous accumulation of these *P*-element variants in these *OreR* strains. Full-length *P*-elements were also sequenced from *Har*, *pi[2]* and *Lerik-P* strains, but there

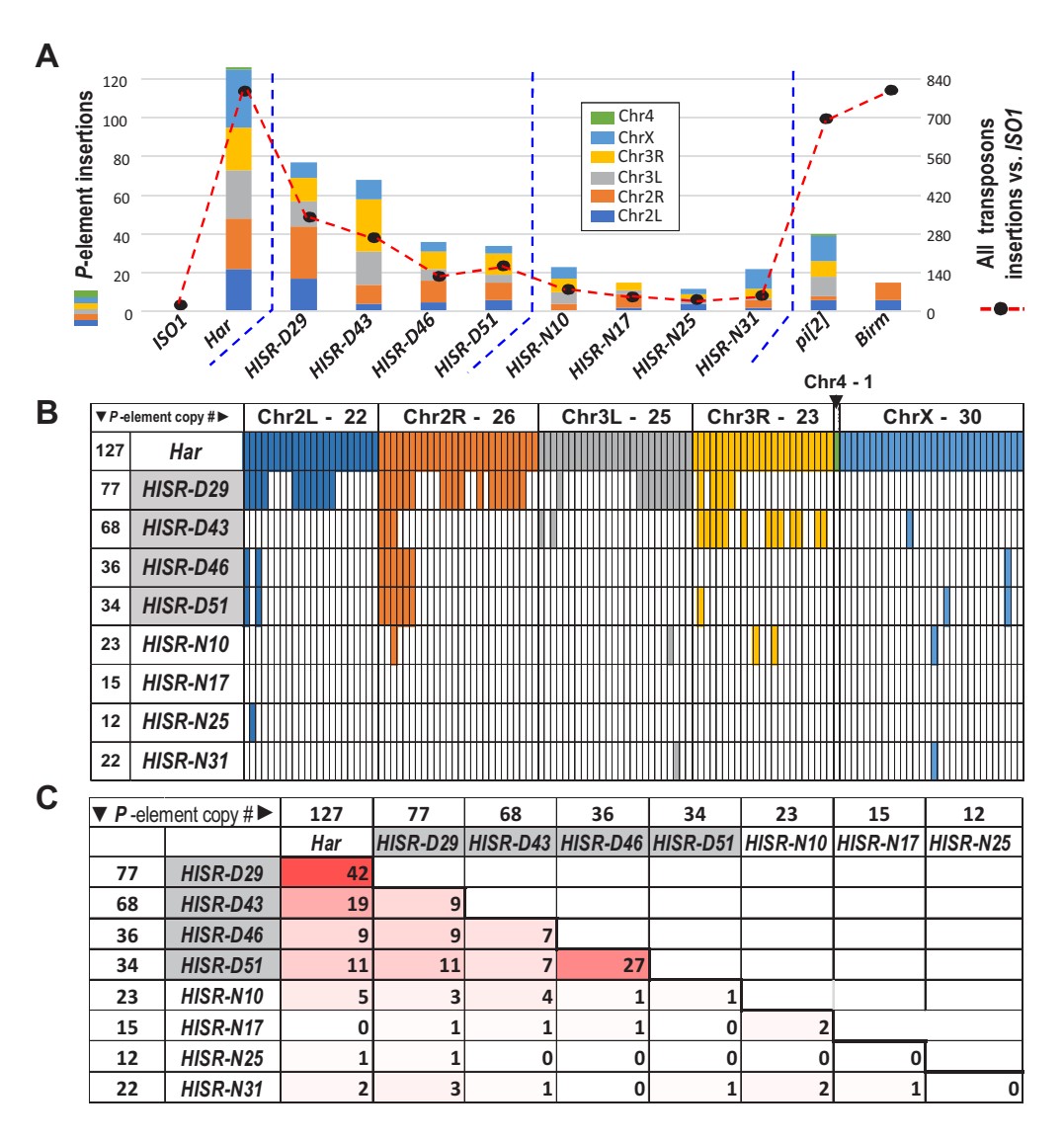

**Figure 4.** *P-elements* are mobilized de novo during generation of *HISR* lines. (**A**) TIDAL program counts of novel *P*-element insertions, left Y-axis and colored bars. Right Y-axis, black dots and dashed line are the total distinct transposon insertions in the unique-mapping portion of genome. (**B**) Lineage analysis of the *Har P*-elements retained in the *HISR* lines, colored by the major chromosomal segments. (**C**) Comparison of shared *P*-elements between *Har* and *HISR* lines, with total number of *P*-elements called by TIDAL in the top row and first column. Color shade reflects degree of shared *P*-elements between the two strains being compared.

The online version of this article includes the following source data and figure supplement(s) for figure 4:

**Source data 1.** Spreadsheets with the calculations of P-element locus comparisons amongst the HISR strains and the records of the de novo P-element insertions amongst the HISR strains.

**Figure supplement 1.** Genome-wide SNP profiling of *HISR* strains.

were no sequence differences in these clones from the original full-length *P*-element sequence in GenBank that might suggest a superlative quality to the full-length *P*-element in these strong GD-inducing strains.

Interestingly, we sequenced short ~630 bp *P*-element variants that were all very similar in configuration in *Har*, *pi[2]*, and *Birm* strains, which only retains ~130 bp of the 5' end and ~500 bp of the 3' end of the *P*-element (**Figure 5A and B**). By retaining functional TIRs, these short elements can still be detected by TIDAL in WGS, can mobilize during crosses with the *pi[2]* strain (**Bingham et al., 1982**; **Mullins et al., 1989**; **O'Hare and Rubin, 1983**); and were previously shown to be able to

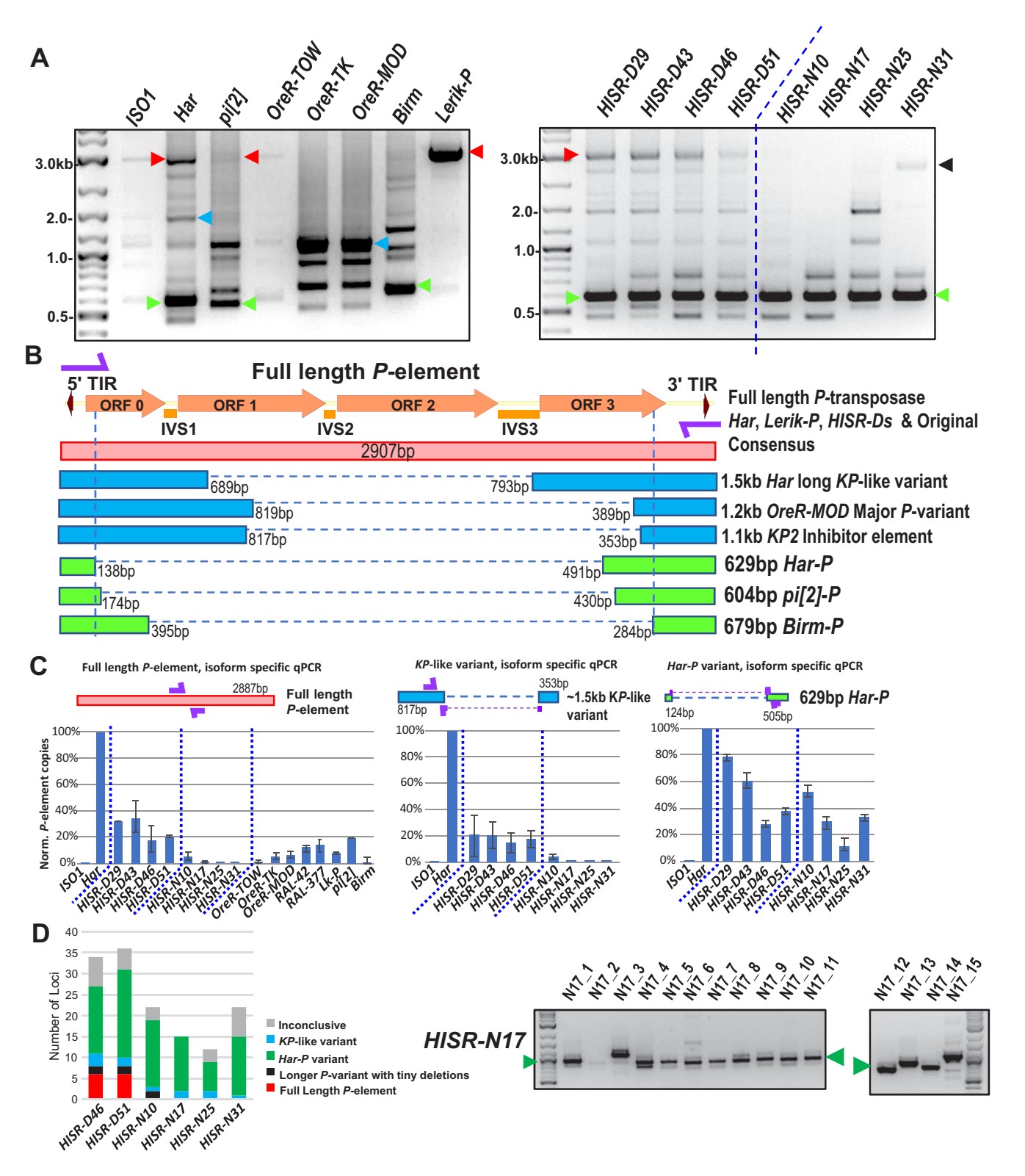

**Figure 5.** *Har-P* is a short and highly mobile *P*-element variant in strains used in *P*-element GD assays. (**A**) *P*-element variant amplicons generated with TIR primers were cloned and sequenced as marked by colored arrows for the sequenced diagrams in (**B**). (**B**) Diagram of *P*-element variants cloned and sequenced from genomic PCR amplicons shown above. (**C**) Genomic qPCR quantifications of three *P*-element variants in *Harwich* and *Harwich*-derived *HISR* lines. Relative quantifications (in percentage) were calculated from ΔΔCt with *rp49* as reference gene. (**D**) Proportions of the *P*-element variants

*Figure 5 continued on next page*

*Figure 5 continued*

verified by locus-specific PCR from TIDAL predictions of all *HISR-N* and *HISR-D* strains with <40 *P*-element insertions. The gel for *HISR-N17* is on the right, while remaining gels are in ***Figure 5—figure supplement 1***.

The online version of this article includes the following source data and figure supplement(s) for figure 5:

**Source data 1.** Spreadsheets with the raw values from the qPCR experiments and calculations of the percentage of the insertions corresponding to the three P-element variant types.

**Figure supplement 1.** Genomic PCR amplifications of *P*-element insertion loci in *HISR-N* and *HISR-D* lines.

**Figure supplement 2.** Further examinations of *P-element* variants amongst distinct *Drosophila* strains that strongly induce GD only when possessing the very short variants similar in length to *Har-P*.

**Figure supplement 2—source data 1.** Spreadsheets with the tabulation of gonadal dysgenesis assays and raw values from the qPCR experiments.

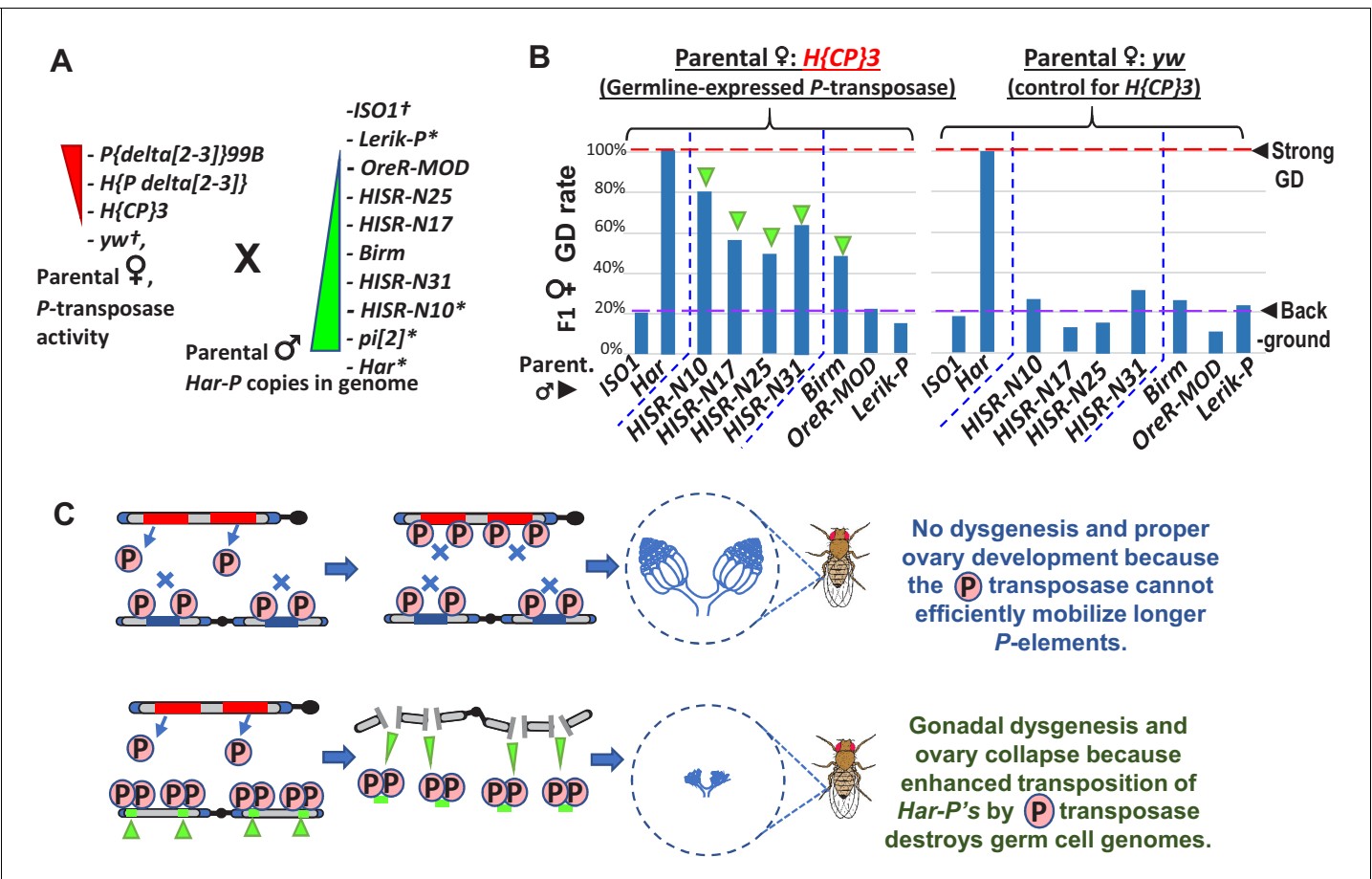

**Figure 6.** In absence of piRNA silencing, *Har-P* crossed with *P*-transposase restores severe GD. (**A**) Strains for testing *Har-P* genetic interaction with *P*-transposase activity. *-strains with piRNA silencing; †-strains lacking any *P*-elements. (**B**) The *H{CP}3* strain's moderately-expressed maternal dose of *P*-transposase crossed with paternal *HISR-N* strains and *Birm* strain restores GD in F1 daughters, but strains with longer and full-length *P*-elements like *OreR-MOD* and *Lerik-P* lack the GD phenotype. (**C**) Model for *P*-transposase mobilizing *Har-Ps* to cause catastrophic transposition.

The online version of this article includes the following source data and figure supplement(s) for figure 6:

**Source data 1.** Spreadsheet with the tabulation of gonadal dysgenesis assays.

**Figure supplement 1.** *P*-transposase RNA expression and *P*-copy number load is largely similar between Dysgenic and Non-dysgenic cross daughters.

**Figure supplement 1—source data 1.** Spreadsheets with the calculations of P-element RNA levels in progeny of crosses and droplet digital PCR counts of P-element copy numbers in progeny ovaries and carcasses.

assemble in vitro with the *P*-transposase tetramer complex >100X more efficiently than the full-length *P*-element (*Tang et al., 2007*). In addition, these short *P*-element variants seemed unlikely to translate into a protein due to multiple premature stop codons introduced by the massive internal deletion.

In all *HISR-D* lines that retain strong GD induction, we detected this short *P*-element variant and the full-length *P*-element encoding *P*-transposase, whereas the *HISR-N* lines retained the short variant but appeared to have lost the full-length *P*-element (*Figure 5A*, right panel). With the smaller number of TIDAL-predicted *P*-element insertions in *HISR-N* lines, we confirmed by locus-specific PCR the absence of full-length *P*-elements and that the majority of *P*-element insertions (~55–95%) were these de novo short *P*-element insertions (*Figure 5D* and *Figure 5—figure supplement 1*). We name this short variant '*Har-Ps*' (*Harwich P's*) in homage to Harpies, highly mobile hybrid bird-human creatures from the Greek mythological stories of the Argonauts.

To further support the conclusion that *Har-P*-like elements are the ammunition to drive severe GD, we examined additional fly lines from the DGRP collection (*Mackay et al., 2012*) which are of completely independent origin from *Har*, *pi[2]*, and *Birm* strains. Indeed, the strains *RAL-42* and *RAL-377* that cause severe GD despite having a fraction of the load of *P*-elements as *Har* also possessed *Har-P-like* short variants (*Figure 5—figure supplement 2*). Meanwhile, two other *P-element*-containing strains *RAL-508* and *RAL-855* did not induce GD because only the longer *KP*-like elements were present (*Figure 5—figure supplement 2*).

## Restoring GD when *Har-P* is crossed with *P*-transposase expressed in the germline

We hypothesized that *Har-Ps* combined with *P*-transposase from full-length *P*-elements could be the drivers of strong GD induction from *pi[2]*, *Har*, and *HISR-D* strains. To test this hypothesis, we used negative-control *yw*-background females that lack *P*-transposase and transgenic *H{CP}3* females that only express *P*-transposase in the germline (*Simmons et al., 2002b*) in crosses with males that either lack *Har-P* copies (*ISO1*, *Lerik-P*, *OreR-MOD*) or contain many *Har-P* copies (*Har*, *HISR-N's*, *Birm*) (*Figure 6A*). GD induction was only restored in the F1 daughters of this cross in strains with many *Har-Ps* (*Figure 6B*). To avoid silencing of *P*-transposase by maternal *P*-element piRNAs in these strains, these crosses specifically used males that should only contribute paternal chromatin without contributing piRNAs (*Figure 6A*). Notably, the *KP*-length and full-length *P*-elements in *OreR-MOD* and *Lerik-P*, respectively, did not restore GD (*Figure 6B*, right most bars of left graph). These results suggest *P*-transposase act upon *Har-P* loci rather than longer *P*-element variants to induce GD and support the observation for *Har-P* loci making up the majority of the de novo *P*-element insertions in *HISR-N* lines (*Figure 6C*). Our data now genetically explain a previously described biochemical result showing that *P*-transposase assembles much more efficiently in vitro on short *P*-elements compared to the full-length *P*-element (*Tang et al., 2007*).

We noticed that GD severity in crossing *HISR-N* with the *H{CP}3* transgenic line was not completely penetrant like GD assays with the parental *Har* (*Figure 6B* versus *Figure 1B*) because *Har* contributes both multiple copies of full-length *P*-elements and *Har-P* loci versus the single copy of the natural *P*-element transgene in *H{CP}3* (*Simmons et al., 2002b*). In addition, natural *P*-element translation is inhibited by strong somatic splicing inhibition of the native *P*-element's third intron (IVS3) containing a premature stop codon and only inefficient splicing in the *Drosophila* germline that is further suppressed by piRNAs (*Siebel et al., 1994*; *Teixeira et al., 2017*). We also confirmed that IVS3 intron splicing was the main alteration that increased *P*-element expression in ovaries from a dysgenic cross between *Har* and *ISO1*, whereas Open Reading Frame (ORF) parts of the *P*-element transcript are only modestly increased (*Figure 6—figure supplement 1A*). We believe this sufficient expression of *P*-transposase promotes the preferred mobilization of *Har-P* short variants in dysgenic cross ovaries, but the cut-and-paste transposition mechanism of *P*-transposase should theoretically conserve the total copy number of *P*-elements. By using digital droplet PCR to precisely quantity total *P*-element copy numbers, we confirmed that total *P*-element copy numbers were stable across ovaries of daughters from two sets of dysgenic and non-dysgenic crosses (*Figure 6—figure supplement 1B*).

## Somatic expression of *P*-transposase with *Har-P's* causes pupal lethality

To test whether a stronger expressing *P*-transposase transgene could induce the complete GD in crosses with *HISR-N* lines, we turned to the *delta[2-3]* *P*-transposase transgenes that lack the IVS3 intron to enable strong somatic and germline *P*-transposase activity (*Robertson et al., 1988*). When we crossed two different *delta[2-3]* female strains to males of *HISR-N17*, *-N25*, and *-N31* which lack *P*-element piRNA expression but have *Har-Ps,* we were unable to assay GD because of extensive pupal lethality (*Figure 7A*). We also confirmed extensive pupal lethality in crosses between *delta[2-3]* and the *Birm* strain (*Figure 7A*) as previously described (*Engels et al., 1987*; *Simmons et al., 1987*). Since we also detected very short *P* variants in *Birm* that are similar to *Har-P* (*Figure 5A and B*) we conclude that somatically expressed *P*-transposase acting only on the *Har-Ps* in *Birm*, *HISR-N17*, *-N25*, and *-N31* is sufficient to disrupt pupal development.

Unexpectedly, the pupal lethality was suppressed when *delta[2-3]* females were crossed with *Har-P*-containing males that also expressed *P*-element piRNAs, such as *Har*, *pi[2]*, the four *HISR-D* lines, and *HISR-N10* (*Figure 7A*). These hybrid F1 progeny developed into adults, but the adult females of *Har* and *HISR-N10* hybrids with *delta[2-3]* still exhibited severe GD (*Figure 7B*). In addition, we also observed severe pupal lethality when *delta[2-3]* females were crossed to *RAL-42* but not *RAL-377*, although strong GD was still observed with *RAL-377* (*Figure 7—figure supplement 1*). These *RAL* strains of independent origin from *Har*, *pi[2]*, and *HISR* lines provide convincing support for the conclusion that *P*-element piRNAs impart a paternally-transmitted imprint on *Har-P* loci that resists mobilization with somatically-expressed *P*-transposase and enables development to adulthood. However, this imprint is either erased in ovaries or insufficient to prevent ovarian GD. Finally, the notable *P*-element piRNA pattern of *HISR-N10* perfectly matches the *Har-P* structure since many internal piRNAs are absent (*Figure 3B*), but overall *P*-element piRNAs in *HISR-N10* are equivalent to *Har* and *HISR-D* lines (*Figure 3C*), and therefore are sufficient to repress *Har-P's* epigenetically from being mobilized in the soma by the *delta[2-3]* *P*-transposase.

## Discussion

After a *Drosophila* strain has silenced an invading transposon through the Piwi/piRNA pathway, the neutered transposon will naturally decay into various truncations that are presumed to be neutral or even beneficial to host fitness (*Kelleher, 2016*), such as natural *KP2* truncation variants that inhibit *P*-element transposition (*Jackson et al., 1988*; *Simmons et al., 1990*). However, we discovered one such truncation we call *Har-P* via our unbiased genetic and molecular approach that can actually be detrimental to the host. Our findings resonate with the previous finding that *P*-transposase assembles in vitro much more efficiently on very short natural *P*-element variants (*Tang et al., 2007*), therefore we propose a new model for catastrophic *P*-element transposition in strong GD inducer strains like *pi[2]* and *Har* (*Figure 6C*).

When a *P*-element truncates to a ~630 bp *Har-P* variant, this non-autonomous variant dominates as the main mobilizing *P*-element during a dysgenic cross to induce strong GD. Thus, previous studies examining GD variability across other *Drosophila* strains and isolates may now be explained by whether these genomes contain both full length and very short *P*-elements (*Bergman et al., 2017*; *Kozeretska et al., 2018*; *Ronsseray et al., 1989*; *Srivastav and Kelleher, 2017*; *Wakisaka et al., 2017*; *Yoshitake et al., 2018*). Moreover, this particular deletion size of a ~630 bp *P*-element variant that arose in at least four completely independent lines has persisted without detriment to these animals either because of piRNA silencing (i.e. *Har*, *pi[2]*, *RAL-377*) or because the *P*-transposase was separated (i.e. *HISR-N* and *Birm*). The short configuration must be special because additional sequence lengths such as *P*-element-based transgenes that were mobilized by *P*-transposase into transgenic strains are not strong triggers of GD like the *Har-P* elements (*Figure 7—figure supplement 1C*).

Although our future goal will be to determine which specific epigenetic marks are deposited at full length *P*-elements and *Har-P's* by piRNAs, we believe a chromatin mark resisting *P*-transposase activity is more likely than somatic piRNAs or siRNAs (*Chung et al., 2008*; *Ghildiyal et al., 2008*; *Kawamura et al., 2008*) silencing the *delta[2-3]* *P*-transposase in our pupal lethality crosses because we confirmed robust *P*-transposase mRNA expression regardless of the expression of *P*-element piRNAs (*Figure 7—figure supplement 2A*). A second future goal will be to generate transgenic flies with single or multiple synthetic *Har-P* copies to determine the precise dosage of *Har-P's* that would

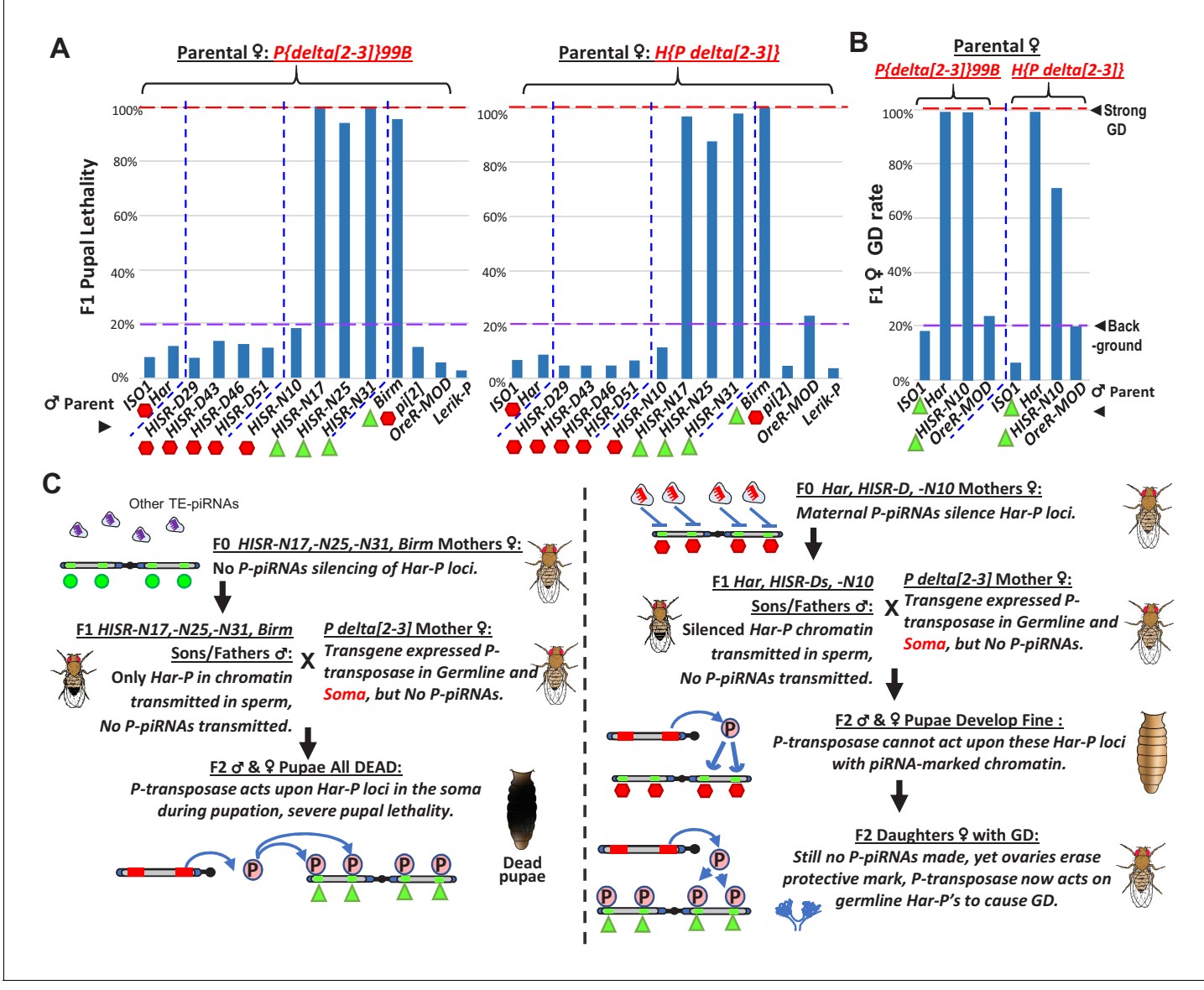

**Figure 7.** Somatic expression of *P*-transposase triggers pupal lethality with *Har-P* loci that are not silenced by *P-element* piRNAs. (A) Green triangles denote crosses showing pupal lethality from stronger somatic expression of *P*-transposase acting on *Har-Ps* in *HISR-N* and *Birm* strains lacking *P*-element piRNAs. Red hexagons denote crosses with strains expressing *P*-element piRNAs that suppress pupal lethality through a paternally transmitted epigenetic imprint. (B) The paternal *P*-element piRNA imprinting on *Har-Ps* in *Har* and *HISR-N10* cannot suppress GD in F1 daughters, as marked by green triangles. The longer *P* variants in *OreR-MOD* do not result in GD with the *delta[2-3]* *P*-transposase. (C) Revised *P* Dysgenesis paradigm proposing a paternally-transmitted piRNA-directed epigenetic mark that resists *P*-transposase activity in the soma, but this mark is erased during oogenesis.

The online version of this article includes the following source data and figure supplement(s) for figure 7:

**Source data 1.** Spreadsheets with the tabulation of pupal lethality and gonadal dysgenesis assays.
**Figure supplement 1.** Further tests of somatic pupal lethality and GD in crosses between *RAL* lines and *delta[2-3]* *P*-transposase lines.
**Figure supplement 1—source data 2.** Spreadsheets with the tabulation of pupal lethality and gonadal dysgenesis assays.
**Figure supplement 2.** Further examinations of *delta[2-3]* *P*-transposase RNA expression and *delta[2-3]* *P*-transposase capacity to cause pupal lethality with lower copy numbers of *Har-P's*.
**Figure supplement 2—source data 1.** Spreadsheets with the raw values from the qPCR experiments and the tabulation of pupal lethality.

trigger GD or pupal lethality. However, in addition to copy number, genomic location may also influence host tolerance of *Har-P's*, because we observed a significant rescue of viable pupae in crosses between *delta[2-3] P*-transposase and a derivative strain of *HISR-N17* with *Har-P's* only on Chromosome 3 with six *P*-elements, while no pupae survived with *delta[2-3] P*-transposase and *Har-P's* on Chromosome 2 with nine *P*-elements (*Figure 7—figure supplement 2B*).

The *Drosophila P*-element system of hybrid GD mainly affects female sterility and requires maternally contributed *P*-element piRNAs to propagate transgenerational *P*-element silencing in daughters via trimethylation of histone H3 lysine 9 (H3K9me3) (*Josse et al., 2007*; *Le Thomas et al., 2014*). Although previous studies of dysgenic crosses focused on complete GD in females (*Bingham et al., 1982*; *Brennecke et al., 2008*; *Khurana et al., 2011*; *Rubin et al., 1982*), sons respond differently because they are fertile despite presumed somatic *P*-element excision (*Wei et al., 1991*). Since previous studies of *P-M* hybrid dysgenesis above never considered a paternal imprint on *P*-elements that our findings now suggest is being propagated (*Figure 7C*), future studies with *HISR-N* strains will enable us to dissect a paternally-transmitted small RNA-directed silencing effect in *Drosophila* that harkens to also a similar paternally-transmitted RNAi effects observed earlier in nematodes (*Grishok et al., 2000*).

Mouse piRNAs bound by MIWI2 direct the re-establishment of DNA methylation marks on transposons like *L1* and *IAP* (*Aravin et al., 2008*), which may propagate in sperm, but *Drosophila* DNA methylation is not prominent (*Krauss and Reuter, 2011*). Similar to other metazoans, *Drosophila* sperm also undergoes histone exchange with protamines (ProtB), with little contribution of paternal cytoplasm (*Rathke et al., 2007*). However, recent data do support the retention of some H3K9me3 in sperm (*Yamaguchi et al., 2018*), which might underlie the paternal imprint of piRNA-silencing of *P*-elements that will be investigated further in future studies.

This interplay between the truncated *Har-Ps* and full-length *P*-element DNA transposon resembles other examples in nature, such as the extreme proliferation of MITEs (miniature inverted repeat transposable element) in rice scavenging other transposases to mobilize (*Yang et al., 2009*), and short mammalian SINEs retrotransposons taking advantage of the transposition machinery of longer LINEs, since SINEs persist in much greater numbers than their longer LINE counterparts (*Hancks and Kazazian, 2012*). However, while the full impact of MITEs and SINEs on organism development is still obscure, our study indicates that *Har-Ps* combined with the *P*-transposase to trigger transposition events so efficiently to be detrimental to ovarian and pupal development. Notwithstanding, the high efficiency of *Har-P* mobilization *by P*-transposase may also be engineered into a new generation of transposon-based mutagenesis approaches.

# Materials and methods

## Key resources table

| Reagent type (species) or resource | Designation | Source reference | Identifier | Additional information |
|---|---|---|---|---|
| Strain, strain background (*Drosophila melanogaster*) | *ISO1* | Susan Celnicker Lab | PMID: 25589440 | iso-1 : y[1]; Gr22b[1] Gr22d[1] cn[1] CG33964[R4.2] bw[1] sp[1]; LysC[1] MstProx[1] GstD5[1] Rh6[1] |
| Strain, strain background (*Drosophila melanogaster*) | *Harwich* | William Theurkauf Lab | PMID: 22196730 | |
| Strain, strain background (*Drosophila melanogaster*) | *HISR-D29* | Created in Lau Lab | This paper | |
| Strain, strain background (*Drosophila melanogaster*) | *HISR-D43* | Created in Lau Lab | This paper | |
| Strain, strain background (*Drosophila melanogaster*) | *HISR-D46* | Created in Lau Lab | This paper | |
| Strain, strain background (*Drosophila melanogaster*) | *HISR-D51* | Created in Lau Lab | This paper | |

*Continued on next page*

*Continued*

| Reagent type (species) or resource | Designation | Source reference | Identifier | Additional information |
|---|---|---|---|---|
| Strain, strain background (*Drosophila melanogaster*) | *HISR-N10* | Created in Lau Lab | This paper | |
| Strain, strain background (*Drosophila melanogaster*) | *HISR-N17* | Created in Lau Lab | This paper | |
| Strain, strain background (*Drosophila melanogaster*) | *HISR-N25* | Created in Lau Lab | This paper | |
| Strain, strain background (*Drosophila melanogaster*) | *HISR-N31* | Created in Lau Lab | This paper | |
| Strain, strain background (*Drosophila melanogaster*) | *H{KP}* | BDSC | Stock No. 64175 | y[1] w[67c23]; H{w[+mC]=KP}H |
| Strain, strain background (*Drosophila melanogaster*) | *pi[2]* | BDSC | Stock No.2384 | |
| Strain, strain background (*Drosophila melanogaster*) | *OreR-MOD* | BDSC | Stock No.25211 | |
| Strain, strain background (*Drosophila melanogaster*) | *OreR-TK* | BDSC | Stock No.2376 | |
| Strain, strain background (*Drosophila melanogaster*) | *OreR-TOW* | Terry Orr-Weaver's Lab | PMID: 21177974 | |
| Strain, strain background (*Drosophila melanogaster*) | *RAL-42* | BDSC | Stock No.28127 | |
| Strain, strain background (*Drosophila melanogaster*) | *RAL-377* | BDSC | Stock No.28186 | |
| Strain, strain background (*Drosophila melanogaster*) | *RAL-508* | BDSC | Stock No.28205 | |
| Strain, strain background (*Drosophila melanogaster*) | RAL-855 | BDSC | Stock No.28251 | |
| Strain, strain background (*Drosophila melanogaster*) | *H{CP}3* | BDSC | Stock No.64160 | y[1] w[67c23]; H{w[+mC]=hsp/CP}3 PMID: 12019234 |
| Strain, strain background (*Drosophila melanogaster*) | *yw* | John Abrams's Lab | PMID: 26701264 | |
| Strain, strain background (*Drosophila melanogaster*) | *P{delta[2-3]}99B* | BDSC | Stock No.3629 | w[*]; wg[Sp-1]/CyO; ry[506] Sb[1] P{ry[+t7.2]=Delta2-3}99B/TM6B, Tb[+] PMID: 3000622 |
| Strain, strain background (*Drosophila melanogaster*) | *H{P delta[2-3]}* | BDSC | Stock No.64161 | y[1] w[67c23]; H{w[+mC]=w[+].Delta2-3.M}6 |
| Strain, strain background (*Drosophila melanogaster*) | *P{TubGal80}Chr2* | BDSC | Stock No.7108 | |
| Strain, strain background (*Drosophila melanogaster*) | *P{TubGal80}Chr3* | BDSC | Stock No.7017 | |
| Strain, strain background (*Drosophila melanogaster*) | *P{Elav-Gal4}Chr2* | BDSC | Stock No.8765 | |
| Strain, strain background (*Drosophila melanogaster*) | *Sp/CyO;TM6b/Sb* | Michael Rosbash's Lab | | |
| Strain, strain background (*Drosophila melanogaster*) | *Lerik-P* | Stephane Ronsseray's Lab | PMID: 1660427 | Lk-P(1A)-SL2 |
| Strain, strain background (*Drosophila melanogaster*) | *Birm* | BDSC | Stock No. 2359 | Birm Chr2; PMID: 2835286 |
| software, algorithm | QuantaSoft Analysis Pro | Bio-Rad | | |

*Continued on next page*

*Continued*

| Reagent type (species) or resource | Designation | Source reference | Identifier | Additional information |
|---|---|---|---|---|
| software, algorithm | Applied Biosystems 7500/7500 Fast Real-Time PCR System v2.0 | Applied Biosystems | | |
| software, algorithm | BWA MEM | *Li and Durbin, 2010* | PMID: 20080505 | |
| software, algorithm | TIDAL-Fly | *Rahman et al., 2015* | PMID: 26578579 | |
| software, algorithm | GATK | *McKenna et al., 2010* | PMID: 20644199 | |
| commerical assay or kit | TOPO PCR Cloning Kit | ThermoFischerSci | Cat No. 450031 | |
| commerical assay or kit | Luna Universal qPCR master mix | New England Biolabs Inc | Cat No. M3003 | |
| commerical assay or kit | NEB Ultra II FS DNA library prep | New England Biolabs Inc | Cat No. E7805 | |
| commerical assay or kit | First Strand cDNA Synthesis Kit using ProtoScript II | New England Biolabs Inc | Cat No. M0368 | |
| commerical assay or kit | NEBNext Small RNA Library Prep Set for Illumina | New England Biolabs Inc | Cat No. E7330 | |
| commerical assay or kit | QX200 ddPCR EvaGreen Supermix | Bio-Rad | Cat No. 1864034 | |
| Chemical compound, drug | Tri-reagent | Molecular Research Center Inc,OH | | |
| Chemical compound, drug | Q Sepharose Fast Flow, 300 mL | GE HealthCare | Cat No. 17051001 | |

## Fly strains

All strains were maintained on standard cornmeal medium at 22°C. Because the *ISO1*(BDSC#2057) stock had accumulated >180 new transposon insertions relative to the original stock sequenced in the Berkeley *Drosophila* genome project (*Adams et al., 2000*; *Rahman et al., 2015*), we obtained the *ISO1* strain from Susan Celniker's lab (*ISO1-SC*). The *Har* strain was obtained from (*Har-WET*) was obtained from the William Theurkauf's lab (*Khurana et al., 2011*). Three *Oregon-R* strains were obtained from Terry Orr-Weaver's lab, *OreR-TOW*, *OreR*-TK (Kaufman, BDSC#2376) and *OreR-MOD* (BDSC#25211). The *Lerik-P* strain was obtained from Stephane Ronserray's lab (*Josse et al., 2007*; *Marin et al., 2000*). All the following strains were also directly obtained from the BDSC – *RAL-42* (#28127), *RAL-377* (#28186), *RAL-508* (#28205), *RAL-855* (#28251), *pi[2]* (#2384), *y[1] w[67c23]; H{w [+mC]=hsp/CP}3* (#64160), *Birmingham; Sb[1]/TM6* (#2539), *w[*]; wg[Sp-1]/CyO; ry[506] Sb[1] P{ry [+t7.2]=Delta2-3}99B/TM6B, Tb[+]* (#3629), *y[1] w[67c23]; H{w[+mC]=w[+].Delta2-3.M}6* (#64161), *H {w[+mC]=KP}* (#64175), *P{TubGal80}Chr2* (#7108), *P{TubGal80}Chr3* (#7017), and *P{Elav-Gal4}Chr2* (#8765). *Sp/CyO;TM6b/Sb* was obtained from Michael Rosbash's lab.

## Crosses, gonadal dysgenesis and pupal lethality assays

All crosses were set up with 3–5 virgin females and 2–4 young males per replicate on standard cornmeal medium at 25°C and parents were purged after 5 days of egg laying (*Srivastav and Kelleher, 2017*). For GD assays, F1 females aged to 4–5 days at 25°C were examined for GD using food dye and GD % shown is average of 3 replicate crosses with total minimum of 100 F1 females assayed (*Simmons et al., 2007*). Somatic pupal lethality was recorded by counting dead (uneclosed) and empty pupae (eclosed) 6 days after first eclosion was observed in respective control cross (*P{delta[2-3]}99B x ISO1* or *H{P delta[2-3]} x ISO1*) (*Engels et al., 1987*). Pupal lethality percentage shown is average of two or more replicate crosses that obtained at least total of 50 F1 pupae each.

## Crossing scheme to generate *HISR* lines

The detailed crossing scheme is illustrated in *Figure 2—figure supplement 1*. After a first cross between virgin *Har* females and *ISO1* males, three more backcrosses of virgin *Har/ISO1* hybrid progeny females mated to *ISO1* males were performed and following the progeny with red eyes to select for the *Har* segment segregating with the *cn, bw, sp*, alleles on Chromosome 2R. We hoped that a

particular set of *P*-elements that drive strong GD induction would co-segregate with red eye color. We then performed a 'Validation Cross' with the F4 hybrid males individually mated to *ISO1* females. We screened >100 individual groups of F4 males for their GD induction, where the early-hatching 3 day old daughters were screened via the squash assay for 100% GD. Only the F5 vials showing 100% GD from F4 males crossed to *ISO1* females were kept, and then were allowed to age and self-crossed and propagated in 11 more generations to attempt to create recombining-inbred-lines (RILs).

Selecting only flies with red eyes required purging any flies emerging with the 'white' eyes of *ISO1* and discarding many vials that failed to generate progeny due to genotoxic collapse from inability to silence *P*-element transposition. At the F16 stage, Har/ISO1 Selfed Red (*HISR*) lines males were rescreened in a Validation Cross with *ISO1* females, this time keeping lines that still caused 100% GD and designated as *HISR-D* (Dysgenic) lines. We also selected additional lines that had now lost GD and allowing for >50% of females to generate egg chambers, and these were designated *HISR-N* (Non-dysgenic) lines. We performed 2 rounds of single-sibling pair mating to further inbreed these lines in an attempt to stabilize the genotypes, and we maintained 4 lines of each *HISR-D* and *HISR-N* for true propagation of just the red or cinnabar eyes and speck phenotype.

## Genomic DNA extraction, PCR, quantitative-PCR and Droplet Digital PCR

Genomic DNA was prepared from 10 young female flies by homogenizing tissues with plastic pestle in 300 µL Lysis buffer (10 mM Tris pH-8.0, 100 mM NaCl, 10 mM EDTA, 0.5% SDS, and Proteinase K at 50 µg/ml) and incubated at 65°C overnight followed by treatment of RNase A at 100 µg/ml at 37°C for 30 mins. 200 µL of 0.5M NaCl was added followed by one volume of Phenol:CHCl$_3$:IAA (at 25:24:1) and spun at 14,000 rpm for 10 min to isolate DNA in aqueous phase. Aqueous phase was extracted again with one volume of CHCl$_3$:IAA (at 24:1) and supplemented with one volume of 5M LiCl and incubated at −20°C and then spun at 15,000 rpm for 15 mins to precipitate RNA. Supernatant was isolated and supplemented with 2 volumes of 100% ethanol and incubated in −20°C for 2 hr and then spun at 15,000 rpm for 20 mins. DNA pellet was washed with chilled 70% ethanol and dissolved in nuclease free water. DNA integrity checked (>10 kb) by running 1 µg on 1% agarose gel with EtBr.

Genomic PCR reactions to characterize *P*-element structural variation were set up in 30 µL reactions of 1X NEB GC buffer, 300 µM dNTPs, 0.5M Betaine, 2.5 mM MgCl$_2$, 0.25 µM of IR primer (*Rasmusson et al., 1993*), 1 µL of Phusion polymerase and 50 ng of genomic DNA and cycled at 94°C for 1 min, 62°C for two mins, 72°C for 4 mins for 27 cycles and followed by 72°C for 15 min. Genomic PCR reactions to characterize *P*-element structural variation in *HISR* lines, predicted by TIDAL were also set up similarly using *P*-element insertion locus specific primers. Genomic PCR reactions for genotyping of *HISR-N* lines were set up similarly but cycled at 94°C for 30 s, 60°C for 15 s, 72°C for 30 s for 27 cycles and followed by 72°C for 5 min.

Genomic qPCR experiments were performed in three biological replicates with two 20 µL technical reactions replicates each, using Luna Mastermix (NEB), primers at 0.5 µM and 20 ng of genomic DNA per reaction in real time quantitative PCR. *P*-element load was calculated from $2^{(-\Delta\Delta Ct)}$ normalized to *Har* at 100% and $\Delta Ct$ from *RP49*. All primers used for are listed in *Supplementary file 1*.

For the Droplet Digital PCR (ddPCR), we utilized the Evagreen Mastermix (Biorad) and conducted on a QX500 ddPCR machine with manual setting of droplet signal thresholds. 10–15 pairs of ovaries and corresponding carcass from 4 to 5 day old F1 females were dissected from dysgenic and non-dysgenic crosses of *Har* and *HISR-D51* with *ISO1* strain at 18 °C. DNA was extracted from the ovaries and carcass and quantified using Qubit 2.0 Fluorometer. Digital PCR probe assays were conducted in 40 µL droplet reactions, generated from 25 µL digital PCR reaction and 70 µL droplet oil each. 25 µL digital PCR reactions were set up with BioRad ddPCR probe supermix, *P*-element7a (FAM) and rp49 (HEX) probes each at 250 nM and 200 pg of DNA. Reactions were cycled at 95 °C for 10 mins followed by 95 °C for 30 s and 58 °C for 1 min for 40 cycles, and 98 °C for 10 mins. Copies/µL values were extracted from QuantaSoft (BioRad) software and *P*-element copies per genome were calculated normalized to rp49.

### P-element amplicon cloning and sequencing

P-elements amplified from IR PCR were purified from 1% agarose gel using QIAquick Gel extraction kit and cloned into pCR4-TOPO vector using Zero Blunt TOPO PCR Cloning Kit at RT, followed by transformation of chemically competent DH10β cells, which were then grown on LB plates with 0.05 mg/ml Kanamycin overnight. 5–10 colonies were screened by PCR and two colonies positive for P-element cloned were chosen for plasmid mini-prep and sequenced using M13 forward and reverse primers for all variants in addition to internal primers to complete the sequencing of full-length P-elements.

### Whole genome sequencing, SNP profile analysis, and TIDAL analysis

Genomic DNA libraries were prepared using NEB Ultra II FS kit E7805. Briefly, 500 ng of genomic DNA (>10 kb) was fragmented at 37°C for 12 min, followed by adaptor ligation and loop excision according to kit manual protocol. Size selection was performed with two rounds of AmpureXP beads addition to select for insert size 150–250 bp as per kit manual. Library PCR amplification was also carried out as per manual instructions for six cycles and purified using one round of AmpureXP beads addition at 0.9X volume. Individual barcoded libraries were quantified on NanoDrop and each diluted to 2 nM and then pooled to produce equimolar concentration.

Whole genome sequencing was performed on an Illumina NextSeq 500 with paired-end reads of 75 bp x 75 bp in the Rosbash lab at Brandeis University. Reads were demultiplexed and trimmed by Trimmomatic to remove low quality bases, and then reads were analyzed by the TIDAL program (*Rahman et al., 2015*). TIDAL outputs were sorted for P-element insertions and the insertion coordinates were compared across the *HISR* lines using SQL queries in MS-Access. To calculate the Single Nucleotide Polymorphism (SNP) profiles, paired-end reads were mapped to the Dm6 *ISO1* genome with 'BWA MEM'(*Li and Durbin, 2010*) using default parameters. PCR duplicates are removed with Picard and SNPs are called with GATK HaplotypeCaller (*Danecek et al., 2011*; *DePristo et al., 2011*; *McKenna et al., 2010*). We then generated the nucleotide distribution for each SNP to ensure that there are at least 20 reads supporting each SNP. Then, we created a unified SNP list by using the union of SNPs from all libraries and carefully noted if each SNP is present in each library. The SNP counts were binned by 5 kb segments and converted into a graphical representation as differences between the reference genome and strain/line in *Figure 4—figure supplement 1*.

### Ovary small RNA sequencing and analysis

To remove the 2S rRNA from *Drosophila* ovaries, we adapted a protocol from our previous Q-sepharose beads matrix technique (*Lau et al., 2009*). About 50 ovaries per parental *Har* and *ISO1* strains and *HISR* lines were dissected from young adult females. Ovaries were then lysed in ice cold 500 ul Elution Buffer (20 mM Hepes pH 7.9 (with KOH), 10% glycerol, 400 mM KOAc, 0.2 mM EDTA, 1.5 mM MgCl2, 1.0 mM DTT, 1X Roche Complete EDTA-free Protease Inhibitor Cocktail) using one freeze-thaw cycle and pulverizing with a blue plastic pestle. A 1.5 ml aliquot of Q-Sepharose FF matrix suspension was washed 1X in water, then 3X in Elution buffer, then incubated for 10 min with the ovaries lysate with occasional agitation in cold room. Ribosomal RNA gets bound by the Q-sepharose, while small RNA RNPs remains in the elution buffer. Elution buffer was removed and then subjected to small RNA extraction with the Tri-reagent protocol. The precipitated small RNAs where then converted into Illumina libraries using the NEBNext Small RNA Library Construction kit. One modification we employed during the overnight linker ligation is to supplement the reactions to 12.5% PEG 8000 to reduce the potential sequence biases from T4 RNA ligase activity.

Small RNA libraries were sequenced as 75 bp single end reads on the NextSeq550. Adapters for the small RNA libraries were removed with CutAdapt and then mapped to the *Drosophila* transposon consensus sequences from RepBase and Flybase using Bowtie v1 with up to two mismatches and R plotting scripts as applied in our previous published studies on *Drosophila* piRNAs (*Clark et al., 2017*; *Sytnikova et al., 2014*).

### RT-qPCR analysis of P-element expression in gonadal dysgenesis and pupal lethality

For this assay, 5–10 pairs of ovaries were dissected from 3 to 5 day old F1 females of dysgenic and non-dysgenic cross with *Har* and *ISO1*, as well as with *Har* and *HISR-D46*. RNA was extracted from

such ovaries and integrity checked by running 1 µg RNA at 2% Agarose II gel (Fischer BioReagents). 3 µg was reverse transcribed using Protoscript RT enzyme (NEB) as per manufacturer's protocol and negative RT control was carried out similarly without RT enzyme. 50 ng of cDNA was used for setting up rp49 PCR reactions (as described above) from RT and corresponding negative RT reactions to evaluate DNA contamination. qPCR reactions for *P*-element ORF2, ORF3, IVS3 were also carried out as genomic qPCR reactions with 20 ng cDNA input and ΔCt were calculated similarly using rp49 RNA levels.

In the RT-qPCR analysis of *H{P delta[2-3]}* gene expression in gonadal dysgenesis and pupal lethality, 5–10 pairs of ovaries and corresponding carcass were dissected from F1 females of pupal lethality crosses conducted at 18 ℃. RNA extraction, reverse transcription, PCR and qPCR reactions were carried out similarly as above. ΔCt were calculated similarly using rp49 RNA levels. Fold change values were obtained from normalizing F1 carcass *P*-element RNA levels to *H{P delta[2-3]}* carcass and F1 ovary *P-element* RNA levels were normalized to *H{P delta[2-3]}* carcass in *Figure 7—figure supplement 2*.

### Isolation of *HISR-N17* autosomes for modulating *Har-P* genomic dosage

*HISR-N17* autosomes were isolated first by crossing virgin *HISR-N17* females with Sp/CyO;TM6b/Sb stock males and using virgin F1 females with CyO and TM6 to cross again with Sp/CyO;TM6b/Sb. F2 males with either *HISR-N17* Chr2 or *HISR-N17* Chr3 were crossed to virgin *H{P delta[2-3]}*. All crosses were performed in triplicates at 25 ℃. F3 pupal lethality was recorded on 16[th] day of the *H{P delta [2-3]}* crosses.

## Acknowledgements

We thank W Theurkauf, S Celniker, S Ronsseray, T Orr-Weaver, and the BDSC (NIH grant P40OD018537) for fly strains; M Rosbash and Brandeis University for deep sequencing and fly food; and D Schwarz, R McCrae, A Grishok and D Cifuentes for comments. SPS and NCL conceived and conducted the experiments, RR and QM provided bioinformatics analyses, JP and SB contributed final experiments, and NCL wrote the paper. This work was supported by NIH grants R01-AG052465 and R21-HD088792 to NCL. Sequencing data are deposited in the NCBI SRA as Study #SRP178563.

## Additional information

### Funding

| Funder | Grant reference number | Author |
| --- | --- | --- |
| National Institutes of Health | R01-AG052465 | Nelson C Lau |
| National Institutes of Health | R21-HD088792 | Nelson C Lau |

The funders had no role in study design, data collection and interpretation, or the decision to submit the work for publication.

### Author contributions

Satyam P Srivastav, Conceptualization, Formal analysis, Validation, Investigation, Visualization, Methodology, Writing—review and editing; Reazur Rahman, Data curation, Software, Formal analysis, Writing—review and editing; Qicheng Ma, Data curation, Software, Formal analysis; Jasmine Pierre, completing follow up experiments for reviewers; Saptaparni Bandyopadhyay, completed follow up experiments for reviewers; Nelson C Lau, Conceptualization, Resources, Data curation, Formal analysis, Supervision, Funding acquisition, Validation, Investigation, Visualization, Methodology, Writing—original draft, Project administration, Writing—review and editing

### Author ORCIDs

Nelson C Lau  https://orcid.org/0000-0001-9907-1404

**Decision letter and Author response**
Decision letter https://doi.org/10.7554/eLife.49948.sa1
Author response https://doi.org/10.7554/eLife.49948.sa2

## Additional files

### Supplementary files
• Supplementary file 1. Oligonucleotides used in this study.

• Supplementary file 2. Whole genome sequencing and small RNA sequencing metadata.

• Transparent reporting form

### Data availability
The high-throughput sequencing data in our study #SRP178563 can be accessed here: http://www.ncbi.nlm.nih.gov/bioproject/?term=PRJNA514796.

The following dataset was generated:

| Author(s) | Year | Dataset title | Dataset URL | Database and Identifier |
|---|---|---|---|---|
| Srivastav SP, Rahman R, Ma Q, Lau NC | 2019 | Har-P is a short P-element variant that collaborates with P transposase to induce strong gonadal dysgenesis | http://www.ncbi.nlm.nih.gov/bioproject/?term=PRJNA514796 | NCBI SRA, SRP178563 |

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
