## [Decision Letter]

**Acceptance summary:**

This paper provides mechanistic insights a long-studied form of hybrid dysgenesis in *Drosophila* in which paternally transmitted *P*-element transposons induce ovarian atrophy when their transposition is not silenced by maternal piRNAs, as occurs when the mothers do not contain *P*-elements. It was previously observed that there is variation in the extent to which different paternal *P*-element containing strains induce this gonadal dysgenesis (GD). This paper presents a variety of data that suggest that the GD-triggering transposition events typically involve truncated forms of the *P*-element mobilized by transposase expressed from intact *P*-elements. This kind of relationship between truncated and intact transposons is common, occurring in plants (MITEs) and mammals (SINEs/LINEs), but is typically viewed as a parasites of parasites relationship. This work reconciles previous work on the differential strength of GD induction, and, more generally, demonstrates that this form of mobilization of non-protein coding transposons can have important functional consequences. The authors also offer some indirect evidence for paternal imprinting on *P*-elements that potentially explains the lack of lethality and male fertility in these crosses, although more work is needed to validate this hypothesis and understand its mechanism. The paper should be interesting to researchers interested in transposons and transposon silencing as well as those studying hybrid inviability and speciation.

**Decision letter after peer review:**

Thank you for submitting your article "*Har-P*, a short *P*-element variant, weaponizes *P-*transposase to severely impair *Drosophila* development" for consideration by *eLife*. Your article has been reviewed by two peer reviewers, and the evaluation has been overseen by Michael Eisen acting as Reviewing Editor. The reviewers have opted to remain anonymous.

The paper stimulated a lively discussion between the reviewers and editors, who all found the work to be intriguing and the experiments well-executed. But, as discussed below, the reviewers had several concerns about the significance and novelty of the findings that will need to be addressed if the paper is to be published in *eLife*.

The manuscript by Sricastav et al. focuses on the serendipitous identification in lines derived from a wild *P*-element-containing *Drosophila* strain (Harwich) of a small *P*-element derivative, *Har-P*, which is efficiently mobilized in the absence of inhibitory piRNAs, and, the authors assert, plays an important role in gonadal dysgenesis.

One immediate issue with the manuscript was that it was a bit hard to follow the series of experiments and their logic, and this will have to be improved for the manuscript to reach the wide and diverse audience of *eLife*. As we see it the series of experiments and observations are:

1) In an effort to map the basis for the gonadal dysgenesis induced by the Harwich (Har) line of *D. melanogaster*, the authors generated a series of hybrid lines between *Har* and a non-GD inducing line such that the lines were expected to only contain a small fraction of the *Har* genome. The idea was to characterize the GD potential of these lines to zero in on genomic regions responsible for GD.

2) In analyzing these lines "*HISR*" lines, the authors realized that a significant amount of *P*-element mobilization had occurred during their generation, making the initial goal of the experiment no longer possible.

3) However they also realized that a large fraction of the mobilized *P*-elements were actually a short version of a *P*-element, which they name *Har-P*. Both GD inducing (*HISR-D*) and non-GD inducing (*HISR-N*) lines had large numbers of de novo *Har-Ps*. The difference between them appeared to be that the *HISR-D* lines retained intact *P*-elements while the *HISR-N* lines lost them.

4) For reasons that are not clearly outlined in the manuscript, the authors pursued the hypothesis that the *Har-Ps* were responsible for the strong GD induction from the *Har* and *HISR-D* lines, with the difference between the *HISR-D* and *HISR-N* lines being the presence of a full-length transposase to mobilize them.

5) The authors analyzed GD in offspring of crosses of various *Har-P*+ and *Har-P*- males with *P*-element naive females either expressing or not expressing P-transposase in the germline. The presence of *P*-transposase in the germline restored GD in *Har-P*+ lines that do not induce it on their own, consistent with the hypothesis that the mobilization of *Har-P* by *P*-transposase encoded by intact *P*-elements is responsible for GD.

All reviewers note the high technical quality of the experiments and found the individual results to be convincing. We have, however, two broad concerns.

Major concern 1: Are *Har-Ps* responsible for the GD phenotype?

While the model – that, in the presence of *P*-transposase, the small size of *Har-Ps* leads to their preferential mobilization and ultimately to GD – is plausible and consistent with data, the experiments do not definitively establish this. Indeed all of the lines that show GD in the presence of germline *P*-transposase in Figure 6B have other difference to the ones that do not than simply the presence of *Har-Ps*. An alternative model, for example, would be that the proliferation of *Har-Ps* is a consequence of a property of these lines that also leads to GD. We would like to see a better argument, ideally with some data, that points more definitively to the presence of *Har-Ps* as the proximal cause of GD.

Furthermore, we would like to see some additional discussion, again ideally backed by data if it is available, of the properties of *Har-Ps* that make function in this way. Several specific questions came up in this regard. How does *Har-P* compare to the KP or the small *P*-elements on *BIRM2* or *BIRM3*? Does *Har-P* make a protein? Why is *Har-P* better at transposition? The authors speculate that it has to do with pairing of the *P-*element ends, but what features of *Har-P* might be causing this. It has generally been observed that larger *P*-elements (and transposons in general) are mobilized less efficiently. So is it just that *Har-Ps* are small, or is there something in particular about the way they are small that allows them to be predicted mobilize more efficiently?

Major concern 2: The results don't differ from expectation given what was already known about *P*-elements and GD.

The most likely proximate cause of GD in the presence of *P*-transposase and *P*-element DNA is double stranded breaks occurring at the ends of the *P*-element (at the TIRs). As *Har-Ps* contain the TIRs, it's expected that they would increase GD severity: the more TIRs that there are, the more double stranded breaks, as long as there is some active transpose around to recognize them and perform the DSB. The fact that the transpose assembles more efficiently on short elements (as pointed out in the manuscript) probably amplifies the effect of these TIRs.

In fact, the copies of TIRs are expected to be more important that the copies of full-length *P*, as it is known already that very little protein is needed to cause dysgenesis. In fact, as the N and D versions of the lines differ primarily in the presence of full-length *P-*element content than in *Har-P* (Figure 5C), the results more clearly confirm the importance of full-length *P*-element in GD.

In short, the results of this experiment seem more or less consistent with what would have been expected given what is known about *P-*elements and GD. That is, the results seem confirmatory rather than surprising, as is portrayed in the paper. We are, however, open to being convinced that we have missed something here.

---

## [Author Response]

[…] Major concern 1: Are Har-Ps responsible for the GD phenotype?While the model – that, in the presence of P-transposase, the small size of Har-Ps leads to their preferential mobilization and ultimately to GD – is plausible and consistent with data, the experiments do not definitively establish this. Indeed all of the lines that show GD in the presence of germline P-transposase in Figure 6B have other difference to the ones that do not than simply the presence of Har-Ps. An alternative model, for example, would be that the proliferation of Har-Ps is a consequence of a property of these lines that also leads to GD. We would like to see a better argument, ideally with some data, that points more definitively to the presence of Har-Ps as the proximal cause of GD.

We thank the reviewers for raising this concern. We are confident that the very short *P-*element variant is the most likely ammunition that P-transposase acts upon to drive the severest forms of GD. Although the *HISR-D* and HISR-N strains are related to the parental *Har* only in terms of de novo mobilized *P-*elements, the genomic location patterns of these de novo *P-*elements are all distinct from each other (Figure 4D). In addition, the large majority of the genomic background of *HISR* strains is *ISO1*, thereby mitigating the concern of an additional property besides the *Har-Ps* that might be lurking amongst the different strains’ genomic background.

Nevertheless, we have now added some new data and text pointing out that our examination of multiple strains that are completely independent in origin from *Har*. In Figure 1 and Figure 5—figure supplement 2, we point to *pi[2]*, *RAL-42*, and *RAL-377* that each have short *P*-variants like *Har-P*, and these strains all cause very severe GD like Har. In contrast, *OreR-TK*, *OreR-MOD*, *RAL-508*, and *RAL-855* all with the longer *KP*-variants and *Lerik-P* with two full-length *P-*elements; these all do not cause severe GD. The new text describing Figure 5—figure supplement 2 is in the last paragraph of the subsection “Dispersed *P-*element landscapes indicate de novo transposition in *HISR* lines”.

In total, our conclusion is well supported by our data, given the multiple tests we have performed with multiple strains of completely independent origins all fitting with the pattern that very short *Har-P* like variants are present in strains that cause strong GD.

Furthermore, we would like to see some additional discussion, again ideally backed by data if it is available, of the properties of Har-Ps that make function in this way. Several specific questions came up in this regard. How does Har-P compare to the KP or the small P-elements on BIRM2 or BIRM3? Does Har-P make a protein?

We thank the reviewers for this question, and we wish to point out that Figure 5B indeed had diagrammed the cloning and sequencing analysis of the short *P-*element variant from the *Birm* strain, which contains both the *Birm2* (chr2) and *Birm3* (chr3) backgrounds and is the only available *Birm* strain from the BDSC fly stock center. These *Birm P-*element variants are only slightly longer than *Har-P* and the *pi[2]-P* variants at 679bp vs. 629bp and 609bp, respectively. But since *Birm* is a completely independent strain from *Har* and the *HISR-N* strains, yet has the same capacity as *HISR-N* strain to contribute to increased GD in crosses with *P*-transposase (Figure 6B), this is another example strongly supporting the model for the short P variants serving as the genomic ammunition to weaponize *P*-transposase to cause GD.

However, to our surprise, we found all the *Har-P* variants we cloned have the minimum overall length on 5’ and 3’ end required for transposition (Mullins et al., 1987) that include all the sites functional transposase interacts. Hence, we propose *Har-Ps* in Harwich, *HISRs*, *pi[2]*, *Birm*, and *RAL-42*, and *RAL-377* drive GD by higher transposition rates, not by its protein product. This data is now described in the subsection “Dispersed *P-*element landscapes indicate de novo transposition in *HISR* lines” and in new Figure 5—figure supplement 2.

It is very unlikely for *Har-P* to encode a protein because the massive deletion removes segments from all 4 ORFs and within 110bp of the start codon is a stop codon, with then many additional subsequent stop codons. We have added the following text: “In addition, these short *P-*element variants seemed unlikely to translate into a protein due to multiple premature stop codons introduced by the massive internal deletion.”

Why is Har-P better at transposition? The authors speculate that it has to do with pairing of the P-element ends, but what features of Har-P might be causing this. It has generally been observed that larger P-elements (and transposons in general) are mobilized less efficiently. So is it just that Har-Ps are small, or is there something in particular about the way they are small that allows them to be predicted mobilize more efficiently?

We thank the reviewers for this question, and wish to highlight sentences in the Introduction and Discussion citing elegant biochemical studies by Donald Rio’s lab in 2007, which used atomic force microscopy to measure recombinant *P*-transposase assembly of a synapsis complex on full-length *P-*element DNA *and* the highly short *P-*element variant from the *pi[2]* strain that is almost the same as the *Har-P* element. The key finding in Tang et al., 2007 is that *P*-transposase transposition complex assembles ~180-fold More Efficiently on the very short *P-*element variant compared to full-length *P-*element. Our genetics and genomics data now provide clear support for how the more efficient assembly of *P*-transposase synapsis complexes on the short variant can actually lead to more efficient transposition in vivo and this hyper transposition activity of the short variant is the driver of severe GD.

Finally, we have added more text to the second paragraph of the Discussion and Figure 7—figure supplement 2C, where we have added additional control crosses with other *P-*element-based transgenes that were mobilized by *P*-transposase into transgenic strains before but are not strong triggers of GD like the *Har-P* elements when re-crossed to *P*-transposase.

Major concern 2: The results don't differ from expectation given what was already known about P-elements and GD.The most likely proximate cause of GD in the presence of P-transposase and P-element DNA is double stranded breaks occurring at the ends of the P-element (at the TIRs). As Har-Ps contain the TIRs, it's expected that they would increase GD severity: the more TIRs that there are, the more double stranded breaks, as long as there is some active transpose around to recognize them and perform the DSB. The fact that the transpose assembles more efficiently on short elements (as pointed out in the manuscript) probably amplifies the effect of these TIRs.In fact, the copies of TIRs are expected to be more important that the copies of full-length P, as it is known already that very little protein is needed to cause dysgenesis. In fact, as the N and D versions of the lines differ primarily in the presence of full-length P-element content than in Har-P (Figure 5C), the results more clearly confirm the importance of full-length P-element in GD.

We thank the reviewers for raising this issue, however the true novelty of our study is to point out that strains containing various longer *P-*element variants (KP and full-length) bearing intact TIRs are poor inducers of GD. Our multiple tests in multiple independent strains shows that only strains bearing very short *Har-P* like variants coupled with a full-length element that can express transposase can induce severe GD, but strains like *Lerik-P* that contains only a full-length *P-*element with TIRs does not induce severe GD.

Since *P-*element structural variants often exhibit internal deletion only, most variants retain TIRs. Hence, several previous studies that found lack of correlation between *P-*element copy and GD, already refuted the ‘more TIR – more GD model’. Due to genetic partitioning of *P-*element variants in *HISRs* and other combination of *P-*element variants in wt strains, we show for the first time the cause of GD variation.

In addition, the *delta[2-3]* strain only expresses *P*-transposase but Lacks TIRs in the *P*-transposase transgene, therefore when this is crossed with *HISR-Ns*, *Birm* and certain *RAL-42* and *RAL-377* strains with very short *P-*elements, severe GD is induced, whereas longer *P-*elements in *OreR* and other *RAL-508* and *RAL-855* strains are incapable of severe GD even when crossed with *delta[2-3]*.

We respectfully assert this is a highly novel finding not only in the *P-*element literature but also in the transposon literature for a DNA element in a metazoan.

In short, the results of this experiment seem more or less consistent with what would have been expected given what is known about P-elements and GD. That is, the results seem confirmatory rather than surprising, as is portrayed in the paper. We are, however, open to being convinced that we have missed something here.

We thank the reviewers for being open to being convinced that they may have missed something here.

Our study is the first to pinpoint that actual sets of *P-*elements that are causative for severe (~100%) GD as observed with Har.

Our study is the first to show that it is the short *P-*elements that when be recombined with *P*-transposase is causative for severe GD, because our data clearly show that other strains with longer but still internally-truncated *P-*elements do Not induce strong GD when recombined with *P*-transposase.

Our study is the first to show genetically that the shortest *P-*element variant mobilizes much more efficiently than the full-length *P-*element.

Our study is the first to show that *P-element* piRNAs in the mother can imprint a silencing mark on *Har-P* loci in sons *that* suppresses somatic *P*-transposase during pupation in the daughters, but then this silencing feature is erased during oogenesis to result in severe GD.

We hope our rebuttal convinces the reviewers to see the merit of this revision that includes new data and can see how our work now advances the field of transposon biology.